# Enhanced Antibacterial Activity through Silver Nanoparticles Deposited onto Carboxylated Graphene Oxide Surface

**DOI:** 10.3390/nano12121949

**Published:** 2022-06-07

**Authors:** Arturo Barjola, María Ángeles Tormo-Mas, Oscar Sahuquillo, Patricia Bernabé-Quispe, José Manuel Pérez, Enrique Giménez

**Affiliations:** 1Instituto de Tecnología de Materiales, Universitat Politècnica de València (UPV), Camino de Vera s/n, 46022 Valencia, Spain; arbarrui@doctor.upv.es (A.B.); ossana@upvnet.upv.es (O.S.); 2Severe Infection Group, Health Research Institute La Fe, University and Polytechnic La Fe Hospital, Avda. Fernando Abril Martorell 106, 46026 Valencia, Spain; patricia_bernabe@iislafe.es (P.B.-Q.); manuel_perez@iislafe.es (J.M.P.)

**Keywords:** carboxylated graphene oxide, antimicrobial, biocide, silver nanoparticles

## Abstract

The strong bactericidal action of silver nanoparticles (AgNPs) is usually limited by their degree of aggregation. Deposition of AgNPs onto a graphene oxide (GO) surface to generate GO-Ag hybrids has been shown to be an effective method of controlling these aggregation problems. In this sense, a novel carboxylated graphene oxide–silver nanoparticle (GOCOOH-Ag) material has been synthesized, and their antibacterial and biofilm formation inhibitions have been studied. AgNPs decorating the GOCOOH surface achieved an average size of 6.74 ± 0.25 nm, which was smaller than that of AgNPs deposited onto the GO surface. In addition, better distribution of AgNPs was achieved using carboxylated material. It is important to highlight the main role of the carboxylic groups in the nucleation and growth of the AgNPs that decorate the GO-based material surface. In vitro antibacterial activity and antibiofilm-forming action were tested against Gram-positive (*Staphylococcus aureus* and *Staphylococcus epidermidis*) and Gram-negative bacteria (*Pseudomonas aeruginosa* and *Escherichia coli*). Both GO-Ag and GOCOOH-Ag reduced bacterial growth, analyzed by time–kill curves. However, the minimum inhibitory concentration and the minimum bactericidal concentration of GOCOOH-Ag were lower than those of GO-Ag for all strains studied, indicating that GOCOOH-Ag has better antibacterial activity. In addition, both nanomaterials prevent biofilm formation, with a higher reduction of biofilm mass and cell viability in the presence of GOCOOH-Ag. The carboxylation functionalization in GO-based materials can be applied to improve the bactericidal and antibiofilm-forming action of the AgNPs.

## 1. Introduction

Infectious diseases caused by multidrug resistant microorganisms have become one of the most concerning threats to human health around the world [1]. The excessive use of antibiotics over the years has led to the rise of drug-resistant pathogens, resulting in poor treatment efficacy and significant economic loss. Alternatively, other antimicrobial agents for whom bacterial resistance is not expected have been used in order to overcome this problem. It is known that various metal ions and metallic oxides present antimicrobial activity against usual bacterial pathogens, although they are not exempt from toxifying normal human cells [2]. Likewise, quaternary ammonium salts have been widely used due to their excellent bactericidal properties [3,4]; however, they can cause resistance to develop in some bacterial strains after prolonged use [5]. On the other hand, topical antiseptics are effective antimicrobial agents for wound treatment, but repeated or inadequate applications may have negative results [6]. Other alternatives, such as pure antimicrobial peptides, have shown promising results; however, they have a high production cost [7].

The advances of nanotechnology in the field of medicine have contributed in recent years to the search for alternatives against microbial resistance to traditional treatments. Due to their large specific surface area and unique properties, many nanoparticles (NPs) show antimicrobial behavior. Metal oxide NPs with photocatalytic activity that are capable of generating reactive oxygen species (ROS), such as TiO [8], CeO [9], and ZnO [10], have been studied. NPs of copper and its oxides represent some of the most explored compounds due to their broad range of bactericidal activity [11]. Nevertheless, silver nanoparticles (AgNPs) are the most used and tested in the field of biomedicine mainly due to their strong cytotoxicity and minimal resistance [12,13,14]. Despite of their outstanding properties, AgNPs also have some drawbacks, such as their tendency to form aggregates due to their high surface energy, which reduces their bactericidal activity [15].

Graphene-based nanomaterials have drawn much attention in the biomedical field due to their unique thermal, chemical, and electric properties [16]. Some of the reported applications are for cellular imaging [17], nanocarriers in drug delivery [18], regenerative medicine [19] or bactericidal agents [20,21,22].

Combinations of graphene oxide (GO) with various biocidal materials, mainly metallic NPs, have been explored, leading to the production of nanocomposites with significantly higher antibacterial activities [3,23,24]. Jizhen Ma et al. [25] modified the surface of GO sheets by depositing AgNPs, and the hybrid material obtained showed an improved bactericidal effect which was attributed by the authors to the synergistic effect produced by combining both materials.

Wangxiao He et al. [26] investigated the bactericidal activity of a TiO_2_NPs-AuNPs-rGO composite against Gram-positive and Gram-negative bacteria and fungus. They found that the ternary composite material showed a higher antibacterial performance than rGO, TiO_2_NPs, and TiO2NPs-rGO did under solar light irradiation for 2 h, which was more effective toward Gram-positive bacteria than Gram-negative bacteria or fungi.

ZnO/GO composites were prepared by Yan-Wen Wang et al. [27], resulting in composites where ZnO NPs remained homogeneously anchored onto GO sheets. GO helped in the dispersion of ZnO NPs and enabled the intimate contact of *Escherichia Coli* with ZnO NPs and zinc ions as well. In addition, the ZnO/GO composites were found to be much less toxic to HeLa cells compared to the equivalent concentration of ZnO NPs in the composites. The results showed that the synergistic effects of GO and ZnO NPs led to the superior antibacterial activity of the composites.

GO has a large specific surface area [28] and tends to form stable colloidal dispersions in water due to the hydrophilicity provided for the oxygen-containing groups, such as hydroxyl, carboxyl, carbonyl, and epoxide, present on their surface [29]. These features make graphene highly biocompatible, thus making it an excellent substrate to generate new materials with bactericidal applications [30,31,32]. Therefore, GO offers many possibilities for tuning their properties through modifications on its oxygenated functional groups. Therefore, increasing the ratio of carboxyl groups distributed on the surface of the graphene oxide sheets makes it possible to achieve a more homogeneous deposition of AgNPs. In turn, the presence of a greater number of carboxyl groups notably improves the dispersion of the material in water while generating more stable dispersions that facilitate its bactericidal action.

In this work, carboxylated GO decorated with AgNPs (GOCOOH-Ag) was synthesized. The structural and morphological features of the obtained materials were studied by combining various techniques, such as scanning electron microscopy (SEM), transmission electron microscopy (TEM), ultraviolet-visible (UV-Vis) spectroscopy, thermogravimetric analysis (TGA), Fourier-transform infrared spectroscopy (FTIR), X-ray diffraction (XRD), and X-ray photoelectron spectroscopy (XPS). Moreover, antibacterial activity and antibiofilm-formation capacity were studied on a Gram+ and Gram- bacteria and further compared to GO-AgNPs (GO-Ag) hybrid material prepared in the same conditions. The results showed that carboxylated material maximizes bactericidal activity and the biofilm inhibition capacity of AgNPs against all strains tested in all ranges of the concentrations evaluated.

## 2. Materials and Methods

### 2.1. Materials

Ultrathin graphite was provided by Avanzare S.L. (La Rioja, Spain). In addition, sulfuric acid (H_2_SO_4_, 95–98%), phosphoric acid (H_3_PO_4_, 85%), potassium permanganate (KMnO_4_), hydrogen peroxide (H_2_O_2_, 30%), absolute ethanol (CH_3_CH_2_OH), chloroacetic acid (ClCH_2_COOH), hydrocloric acid (HCl), sodium hydroxide (NaOH), and reagent grade sodium borohydride (NaBH_4_) were purchased from Alfa Aesar, Sigma-Aldrich (Valencia, Spain) and used as received.

### 2.2. GO Synthesis

GO was synthesized from natural graphite by means of the improved Hummer’s method [33]. A mixture of concentrated H_2_SO_4_/H_3_PO_4_ (9:1 v:v) was added to a mixture of graphite powder and KMnO_4_ (1:6 wt) in an ice bath and was later heated to 50 °C, maintaining this temperature under constant stirring for 12 h. The resulting product was cooled to room temperature and subsequently poured into ice water with H_2_O_2_ (30%). The mixture was sifted through a testing sieve (250 µm) and centrifuged (4000 rpm for 4 h), discarding the supernatant. The remaining solid material was then washed twice with each 200 mL of water, 200 mL of 30% HCl, and 200 mL of ethanol. Once the multiple-wash process was finished, the remaining material was coagulated with 200 mL of ether and filtered with a 0.45 µm pore size membrane. The solid obtained on the filter was collected with deionized water and freeze dried under high vacuum.

### 2.3. GO-COOH Synthesis

Carboxylic acid-functionalized GO nanosheets (GOCOOH) were obtained through reaction with chloroacetic acid under strong basic conditions, in which the oxygen-containing groups of GO were converted to carboxylic groups [34]. Briefly, 1.2 g of Chloroacetic acid and 1 g of NaOH were added to 100 mL of a GO dispersion (0.5 mg/mL). The mixture was bath-sonicated for 3 h at room temperature. The resulting solution was neutralized with HCl and then purified by repeated rinsing and filtration. The product obtained was finally redispersed and lyophilized.

### 2.4. GO-Ag and GOCOOH-Ag Nanocomposites Synthesis

Typically, 50 mg of GO was dispersed in 65 mL of water by ultrasonic bath treatment for 2 h. Then, 80 mg of AgNO_3_ was added in the former dispersion on a magnetic stirrer to be mechanically mixed for 3 h. Next, 35 mL of a freshly prepared 0.1 M NaBH_4_ aqueous solution was added drop by drop into the previously prepared solution and left under magnetic stirring for 6 h. The resulting dispersion was filtered and rinsed several times with deionized water. GO-Ag nanocomposite was finally obtained by freeze drying. GOCOOH was subjected to the same process to get the GOCOOH-Ag nanocomposite. AgNPs were synthesized by the same procedure without the presence of GO or GOCOOH.

### 2.5. Characterization

UV−Vis absorption spectra were measured using JASCO V-670 spectrometer. The XRD measurements were performed on a 2D Phaser equipment (Bruker, Madrid, Spain), with Cu-Kα radiation working at 30 kV and 10 mA, in order to check the crystalline structure of the GO-based materials. Thermogravimetric analysis (TGA) was performed on a TGA Q50 thermogravimetric analyzer (TA Instruments, Cerdanyola del Valles, Spain). Samples (5−10 mg) were weighed in titanium crucibles and heated under a nitrogen atmosphere from 50 to 800 °C at a heating rate of 10 °C min^−1^. Surface morphologies were obtained by SEM (JSM 6300 JEOL, Tokyo, Japan) equipped with energy-dispersive X-ray spectrometer (EDS, Oxford Instruments, Bristol, UK) for elemental composition measurements. TEM images were acquired using JEM-1010 (JEOL DEBEN AMT, Tokyo, Japan). The presence of functional groups was assessed by means of FTIR. FTIR spectra were acquired on a FT/IR-6200 (Jasco, Madrid, Spain) spectrometer in the spectral window of 4000–400 cm^−1^ in ATR mode. The structure and composition of GO-based materials were studied by X-ray photoelectron spectroscopy (XPS) with a VG-Microtech Multilab 3000 (Thermo Fisher Scientific Inc., Waltham, MA, USA) photoelectron spectrometer. Potential zeta was recorded using a dynamic laser scattering analyzer (Zetasizer, 2000 HAS, Malvern, Worcestershire, UK).

### 2.6. Antibacterial Activity

Antibacterial activity was ascertained by determining the minimum inhibitory concentration (MIC), minimum bactericidal concentration (MBC), time–killing curves and biofilm activity. MICs and MBCs were evaluated against four Gram-positive bacteria (*Staphylococcus aureus* V329 [35] and ATCC25423; *Staphylococcus epidermidis* RP62A and ATCC32984) and two Gram-negative bacteria (*Pseudomonas aeruginosa* PFQ2 and *E. coli* ATCC25922). Strains V329, RP62A, PFQ2, and ATCC25922 are biofilm-forming. Prior to analysis, each isolate was subcultured twice onto brain heart infusion (BHI, Scharlau) agar plates to ensure purity and viability of the test organism. Stock inoculum suspension was prepared by suspending 1 to 3 colonies in phosphate-buffered saline with a pH of 7.3 (PBS) from a 24 h culture and adjusted to a cell turbidity of 0.5 McFarland providing 1.5 × 108 CFU/mL. The density of inocula was confirmed by quantitative colony-forming units (CFUs).

Firstly, a Kirby–Bauer plate diffusion method was performed to determine if GO and GOCOOH showed antibacterial activity by plating a 10 µL drop of a 10 mg/mL solution of GO materials onto Mueller–Hinton agar plates previously inoculated with the microorganism. After 24 h of incubation at 37 °C, the growth inhibition zone was measured. All experiments measuring the antibacterial activity had a control that followed the same condition but without GO-Ag or GOCOOH-Ag.

MIC of GO-Ag and GOCOOH-Ag were determined by means of a two-fold broth microdilution method using BHI as the culture medium and a final inoculum of 3 × 106 CFU/mL. The MICs were determined both visually and by spectrophotometer after 24 and 48 h of incubation and were defined as the lowest drug concentration which completely inhibited growth. Concentrations of GO-Ag and GOCOOH-Ag tested ranged from 100 µg/mL to 0.1 µg/mL and were obtained from a stock water solution of the nanohybrid of 200 µg/mL, which was sonicated for 5 min in an ultrasonic water bath. Further, two-fold dilutions were made in BHI. All concentrations of GO-Ag and GOCOOH-Ag are given in terms of Ag.

MBC was determined by transferring 0.1 mL from the clear (no growth) MIC wells onto BHI agar plates. Plates were incubated at 37 °C for 24 h. The MBC was defined as the lowest drug concentration that killed ≥99.9% (≤10 colonies per plate) of bacteria.

Time–kill curves were evaluated in the presence of 12.66 µg/mL of GO-Ag or GOCOOH-Ag in a volume of 2 mL. The inoculum of cultures was set at 1.5 × 105 CFU/mL. Bacteria were incubated at 37 °C with agitation, and the number of CFU at selected time intervals (0, 1, 2, 3, 4, 6, and 24 h) was determined following the track-dilution method [36]. Briefly, aliquots of 0.1 mL of the cell suspension were serially ten-fold diluted up to 10-6 in a saline solution and 10 μL of various dilutions were spotted and track-extended onto BHI agar plates. After 24 h incubation at 37 °C, the number of CFU was determined. For each time and dilution, three replicates were performed. Assays were repeated on three different days.

The ability of GO-Ag and GOCOOH-Ag to prevent biofilm formation was quantified essentially as described elsewhere [37]. Biofilm was formed in a sterile flat-bottomed 96-well polystyrene microtiter plate (Sarstedt). Wells were filled with 200 µL of TSB supplemented with 0.25 wt% glucose (TSBG, Scharlau) and GO-Ag or GOCOOH-Ag at concentrations ranging from 50.62 to 0.1 μg/mL and 20 μL of the stock inoculum suspension, providing 3 × 106 CFU/mL. After 48 h of incubation at 37 °C, the wells were gently washed three times with 200 µL sterile PBS, air-dried in an inverted position, and stained with 0.1% safranin for 30 s. The wells were rinsed and air-dried again, and the quantification of biofilm formed was determined by measuring the absorbance at 450 nm (Synergy H1 microplate reader, BioTek). Each concentration was tested in quadruplicate and on three separate experiments.

The cells’ viability in the biofilm in the presence of GO-based materials was determined by scanning confocal laser microscopy. Biofilms were formed on sterile flat-bottomed 24-well polystyrene plates (Sarstedt, Nümbrecht, Germany) equipped with sterile glass discs. Wells were filled with 1 mL of TSBG and inoculated with 20 μL of the stock inoculum suspension, providing 3 × 106 CFU/mL, and then incubated for 24–48 h at 37 °C. Then, glass discs were washed three times with 500 µL of sterile PBS to remove planktonic cells and stained with the LIVE/DEADTM BacLightTM bacterial viability kit (Thermo Fisher Scientific, Waltham, MA, USA), following the manufacturer’s instructions. This stain contains SYTO^®^9 and propidium iodide that, in a suitable mixture, allow the visualization of viable bacteria (with an intact membrane), stained with green fluorescence, and nonviable bacteria (with a compromised membrane), stained with red fluorescence. Microscopy and imaging were performed with a Leica SP5 confocal microscope (Leica, Wetzlar, Germany) using sequential mode and 40× oil objective. The excitation wavelengths were 480 nm for SYTO^®^ 9 stain and 490 nm for propidium iodide, and emission wavelengths were 500 and 635 nm, respectively. Representative images were selected from at least three distinct regions on the slide.

All experiments of this section were performed in triplicate and on three separate days to ensure reproducibility. Results were expressed as mean ± SD.

## 3. Results and Discussion

### 3.1. Characterization of Materials

A carboxylation procedure was applied to the parent GO with the aim of improving the AgNPs deposition process by achieving a more homogeneous decoration on the surface and avoiding the problems related to its aggregation, which is highly desirable for the bactericidal response of the material.

Morphology of the GO-based nanomaterials decorated with silver nanoparticles was studied by TEM and shown in Figure 1. TEM micrographs (Figure 1a–c) reveal a homogeneous dispersion, with few aggregations, of sphere-like AgNPs decorating the sheets of GO-based materials. On the contrary, the formation of aggregates is observed for AgNPs obtained without the presence of GO or GOCOOH materials (Appendix A). The EDS mapping shown in Figure 1d reveals the presence of AgNPs covering GOCOOH sheets. These results confirm the role of GO in the nucleation process and deposition of AgNPs through strong interactions established between Ag+ ions and the oxygen-containing functional groups on the material surface. In the case of GOCOOH-Ag, the average size of the AgNPs was 6.74 ± 0.25 nm (Appendix A) against 11.69 ± 4.82 nm (Appendix A) found for GO-Ag and 25.73 ± 0.25 nm (Appendix A) calculated for AgNPs. In fact, 86% of the counted AgNPs on GOCOOH sheets were less than 10 nm in diameter. This narrow size distribution can be explained by both the strong reducing effect of borohydride and the stabilizing role of carboxylic groups, which allow the formation of smaller and better-distributed AgNPs for GOCOOH than for GO.

The UV–Vis spectrum of GO shown in Figure 2a exhibits a main absorption peak centered at 230 nm, which corresponds to the electronic π–π* transitions of C-C aromatic bonds, and a shoulder at 300 nm associated to the n–π* transitions of C=O bonds [38]. New bands at 408 nm for GO-Ag and 405 nm for GOCOOH-Ag, corresponding to a surface plasmon of AgNPs, were observed, thus confirming the deposition of AgNPs on the GO-based material surface. Moreover, the symmetrical shape of the UV–Vis absorption peak and its position indicate a relatively narrow size distribution of small silver AgNPs [39].

Figure 2b shows the XRD patterns recorded for GO-Ag and GOCOOH-Ag nanomaterials. The presence of AgNPs on the GO-based nanosheet surface was confirmed by peaks at 2θ values of about 38.1°, 44.3°, 64.5°, and 77.5°, which are assigned to the (1 1 1), (2 0 0), (2 2 0), and (3 1 1) crystallographic planes of face-centered cubic (fcc) AgNPs, respectively [JCPDS card No. 07-0783]. Sharp diffraction peaks at 11.3° for GOCOOH and 10.3° for GO (Appendix A) were indexed to the (002) plane. For both silver-decorated nanomaterials, this peak was slightly shifted to a greater angle, and a shoulder attributed to a hexagonal graphite structure appeared around 22°, indicating that the process was carried out to incorporate AgNPs onto GO-based materials and induce a certain reduction in them, thereby pointing out the partial restoration of the original graphitic order. Consequently, the intersheet distance decreases with silver functionalization. In the case of GO, this value has a greater change than in GOCOOH, from 0.86 to 0.74 Å, when GO is decorated with AgNPs, compared to GOCOOH, where the distance between the adjacent planes decreases from 0.79 to 0.73 Å. Additionally, both GO-Ag and GOCOOH-Ag nanomaterials present a similar intersheet distance of around 0.74 Å.

TGA measurements were conducted to characterize GO and GOCOOH and their AgNP nanohybrids, and the results are shown in Figure 3. All materials showed an initial weight decrease up to 100 °C due to the removal of adsorbed water [40]. This loss of weight is similar in GO and GOCOOH, which are smaller for AgNP-decorated GO-based nanomaterials, because the AgNPs’ deposition process induces a certain degree of reduction on the GO surface, increasing their hydrophobicity [41]. For GO and GOCOOH, oxygen-containing functional groups start their decomposition at 188 and 149 °C, respectively, indicating the greater presence of carboxyl groups on the GOCOOH surface, which tend to decompose at lower temperatures [42]. When the temperature is increased above this point, a weight loss attributed to the decomposition of the more stable oxygen functionalities is observed for both materials [43]. Incorporation of AgNPs enhanced the thermal stability for GO-Ag and GOCOOH-Ag, indicating the participation of the oxygen functional groups in the AgNPs deposition process [44]. Furthermore, the final residual mass is more significant in AgNP-decorated GO-based nanomaterials, particularly in GOCOOH-Ag, which displayed the highest amount, attributed to the remaining GO carbon skeleton as well as to AgNPs [45]. Therefore, the results obtained by TGA showed that carboxyl groups play an important role in the silver-decorating process.

The typical peaks related with the surface functional groups of GOCOOH and GOCOOH-Ag were displayed in the FTIR spectra as shown in Figure 4. The signal corresponding to carboxylic groups (C=O) was recorded at 1716 cm^−1^. The peaks at 3433 cm^−1^, associated with OH-group stretching, and at 1384 cm^−1^, associated with the deformation vibration of C–OH, were observed for both graphene-oxide-based materials studied. The band at 1234 cm^−1^ was associated with the presence of the epoxide groups (C–O–C). Moreover, the C-O stretching appeared at 1058 cm^−1^, and the peak at 1612 cm^−1^ was attributed to the skeletal vibration of graphitic skeleton [46,47]. After functionalization with AgNPs, the peaks corresponding to the oxygen functional groups showed a decrease in intensity, which indicates the participation of the oxygen groups in the silver-reduction and functionalization processes [48]. GO and GO-Ag (Appendix A) showed the same trend, but a broad band in the C-OH-stretching band region was observed for GO-Ag due to some quantity of water molecules absorbed onto their surface.

Figure 5 shows the XPS spectra used to characterize the surface composition of the GO-based materials and their Ag-decorated hybrid materials. The C1s spectrum for GO (Figure 5a) shows several peaks at 284.60, 285.12, 286.84, and 288.49 eV, attributed to the C–C, C=C, C–O, and O–C=O groups, respectively [49]. The intensity of the O–C=O peak increased after the functionalization (Figure 5b), indicating the successful carboxylation of GO [50]. However, the decrease of the C–O-related-peak and the increase of the C=C peak confirmed a slight partial reduction of the GO during the carboxylation process, as seen by XRD analysis.

GO-Ag (Figure 5c) and GOCOOH-Ag (Figure 5d) showed in both cases a clear decrease in the intensity of the signal associated with the carboxyl groups, (i.e., around 288.5 eV). This decrease was more intense for the compound based on GOCOOH than for the one based on GO.

The presence of signals at 368.3 and 374.3 eV due to Ag 3d3/2 and Ag 3d5/2 (Figure 5e,f) suggests the formation of AgNPs onto GO and GOCOOH nanosheets. Moreover, the splitting of the 3d doublet of Ag is 6.0 eV, indicating the formation of metallic silver [51]. The XPS results, along with the above XRD and TEM results, clearly indicate that the AgNPs are well-assembled on GO-based composites.

The results obtained by XPS measurements can be explained by assuming that the carboxylic acid, hydroxyl or epoxide groups on the GO surface can act as nucleation sites for the growth of the AgNPs and their further deposition onto GO sheets. Silver cations can be preferably attached to ionizable carboxylic functionalities, favoring the deposition process onto the exfoliated GOCOOH sheets through NaBH_4_ reduction.

Zeta potential measurements were carried out with the aim to further study the carboxylic acid groups’ role in the AgNPs’ deposition onto GO-based materials as well as to investigate their stability in water. Zeta potentials of GO, GOCOOH, and hybrid solutions GO-Ag and GOCOOH-Ag were recorded at a pH of 6.0. Water suspension of GO exhibited a zeta potential of −39.6 mV similar to that of GOCOOH, whose value was −40.3 mV, due to the negatively charged surfaces caused by the presence of oxygen functional groups such as hydroxyl, epoxide, and carboxyl.

Functionalization of hybrid materials with AgNPs caused a decrease in zeta potential values, yielding −35.9 mV for GO-Ag and −31.9 mV for GOCOOH-Ag. This decrease was more pronounced for the carboxylated product, indicating the participation of these groups in a preferential way in the deposition process of the NPs on the surface of the material [52], thus allowing GOCOOH-Ag to present a higher concentration of AgNPs than GO-Ag. The range of values obtained, all of them negatively charged, indicates a good stability in an aqueous solution for the GO and GOCOOH compounds, slight improvement for the carboxylated material, and a moderate but sufficient stability for the Ag-decorated nanohybrids.

### 3.2. Antibacterial Activity

Although several studies have reported that pristine GO presents antibacterial activity by itself [53,54,55], the Kirby–Bauer diffusion technique showed that neither GO nor GOCOOH exhibited zone inhibition, demonstrating the absence of antibacterial activity (Appendix A). We attribute this fact to the variability of the GO-based materials, which can be influenced by many kinds of factors, such as the original graphite used for the GO synthesis, the different methods followed for the oxidation required to obtain the graphite oxide, the sonication step usually employed to separating the GO stacked sheets into individual sheets, the laborious filtration and drying process, or even the time and conditions in which the material has been stored, which can also alter its properties. On the other hand, GO-based materials can avoid AgNP aggregation. In this sense, many reports have established that AgNPs improve their antibacterial activity when they are deposited onto GO-based materials [25,48,56]. Additionally, the Kirby–Bauer plate diffusion method was also applied for AgNP, (Appendix A) GO-Ag (Appendix A), and GOCOOH-Ag (Appendix A) initial screening. The greater activity of GO-based materials against free AgNPs was confirmed. Therefore, further characterization was carried out with GO-Ag and GOCOOH-Ag materials.

As shown in Table 1, MIC and MBC of GOCOOH-Ag were lower than those of GO-Ag for all strains studied, indicating that GOCOOH-Ag has better antibacterial activity. The lowest 24 h MIC of GOCOOH-Ag was 3.16 µg/mL for *S. aureus* V329, whereas that of GO-Ag in the same conditions was 5.2 times greater. *S. epidermidis* RP62A and *E. coli* ATCC25922 were more resistant, requiring 12.66 µg/mL of GOCOOH-Ag to be inhibited versus 32.92 and 16.46 µg/mL of GO-Ag, respectively. The 48 h MBC/MIC ratios of GOCOOH-Ag were ≤2 for all species tested (except *S. epidermidis* RP62A, which was >2), which is related to bactericidal activity [57]. The MBC/MIC ratio of GO-Ag was >2 for *S. aureus*, suggesting bacteriostatic activity ≤2 for the other strains.

The bactericidal action, determined by time–kill studies, was found to be dependent on both species and nanomaterial. The time–kill curves showed that 12.66 µg/mL of GO-Ag and GOCOOH-Ag reduced the bacterial growth with respect to the growth control of all strains tested. The killing activity of GOCOOH-Ag was very fast against *P. aeruginosa*, killing 100% of cells in 3 h (Figure 6). Against *E. coli*, GOCOOH-Ag required 5 h to kill 99.9% of cells and 6 h against *S. aureus*. However, no killing activity against *S. epidermidis* was observed, and the growth was always under the growth control (Figure 6 and Appendix A). In contrast, with the same concentration of GO-Ag, there was a slight killing (decrease in viable cells) in the first 6 h followed by an increase in viable cells, but always below the control. The greatest CFU reduction at 24 h with respect to the growth control ranged between 1 and 2 Log depending on the strain.

From the MIC, MBC and time–kill study results, it can be concluded that the antimicrobial activity of AgNPs is considerably enhanced when they are loaded onto the GOCOOH surface. Activity of GO-based nanomaterials on biofilm formation was tested against two Gram-positive (V329 and RP62A) and one Gram-negative (PFQ2) bacteria, all of them with a high biofilm-forming capacity, and a Gram-negative strain (ATCC 25922) which presents a low tendency toward biofilm formation.

Figure 7 and Appendix A show the quantification and visualization of biofilm formed and stained with safranin. Both nanomaterials prevent biofilm formation depending on the strain and concentration tested. The highest inhibitions were obtained with GOCOOH-Ag, and the minimum concentration required to completely inhibit biofilm formation ranged between 6.33 and 12.66 µg/mL. By contrast, with GO-Ag the minimum concentration to inhibit biofilm formation was four times higher than that of GOCOOH-Ag for all species except *E. coli* ATCC25922, which was similar for the two nanomaterials. The fact that both nanomaterials showed similar activity against *E. coli* could be related to the lower biofilm-forming capacity of this strain; consequently, a lower AgNP concentration could be required to prevent biofilm formation, showing a minimal concentration to inhibit biofilm formation of 12.66 µg/mL in the presence of GO-Ag.

The viability assay, by means of confocal microscopy performed on the biofilm developing on glass discs in the presence of Ag-decorated GO-based nanomaterials, showed a considerable reduction of both biofilm mass and viable bacteria in all species tested (Figure 8 and Appendix A). Overall, over 6.33 µg/mL of GOCOOH-Ag a total absence of bacteria was observed, confirming no biofilm formation. However, with the same concentration of GO-Ag, only a reduction in biofilm mass and cellular viability with respect to control was achieved.

In the confocal images of controls, different capacities to form biofilm were observed, as were different ratios of live/dead cells, depending on the strain studied. The strains that showed greater thickness in the biofilm layer and, therefore, higher capacity to form biofilm were the strains V329 *S. aureus* and RP62A *S. epidermidis*, which is consistent with the higher OD540nm values obtained in the biofilm tests on microtiter plates (Figure 7 and Appendix A). Therefore, in both strains, significant differences were observed between the biofilm formed in the controls and the biofilm treated with GOCOOH-Ag. Regarding the live/dead cell ratio, in V329 *S. aureus* and ATCC25922 *E. coli* strains, a low proportion of dead cells was observed in the controls, with an increase after treatment with GO-Ag. However, in RP62A *S. epidermidis* and PFQ2 *P. aeruginosa* strains, a greater proportion of dead bacteria was observed in the controls; therefore, the proportion of dead cells after GO-Ag treatment increased less.

## 4. Conclusions

The bactericidal and antibiofilm-forming activity of AgNPs decorating the synthesized GO and GOCOOH materials were evaluated by using five techniques. The activity of AgNPs as a bactericidal and antibiofilm-forming agent was observed in both materials. Furthermore, the efficiency of AgNPs decorating GO greatly improved when they were deposited onto the GOCOOH surface. The lowest 24 h MIC value GOCOOH-Ag was 3.16 µg/mL for *S. aureus* V329, whereas GO-Ag in the same conditions was 5.2 times greater. On the other hand, to be inhibited, *S. epidermidis* RP62A and *E. coli* ATCC25922, the GO-Ag material required 2.6 and 1.3 times greater MIC values than GOCOOH-Ag at 24 h, respectively. Morphological characterization reveals a homogeneous dispersion of AgNPs with a narrow size distribution, achieved by the greater amount of carboxyl groups present on the GOCOOH surface, which can act as nucleation sites for the AgNPs growth. In addition, the prepared GO-based hybrid materials were easily dispersible in water-generating stable dispersions. In summary, the results obtained in this work open the door to exploring new applications of GOCOOH-Ag material as a bactericidal and antibiofilm-forming agent in the field of nanomedicine.

## Figures and Tables

**Figure 1 nanomaterials-12-01949-f001:**
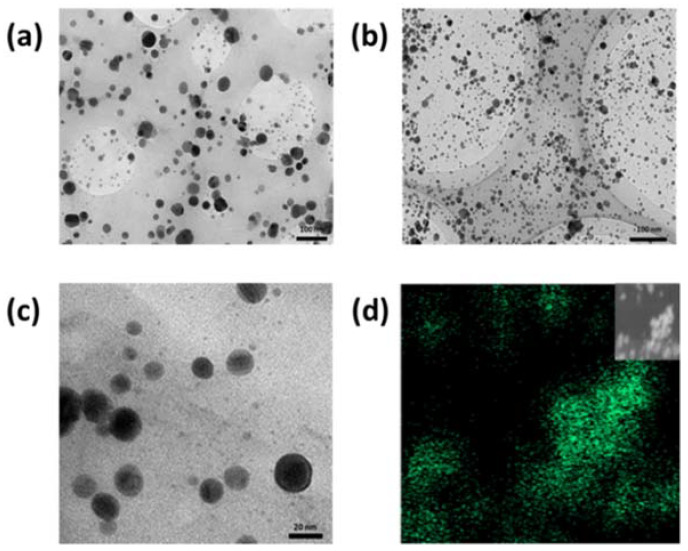
TEM micrographs of (**a**) GO-Ag and (**b**) GOCOOH-Ag, (**c**) magnified image of GOCOOH-Ag, and (**d**) EDS mapping of AgNPs deposited on the GOCOOH surface.

**Figure 2 nanomaterials-12-01949-f002:**
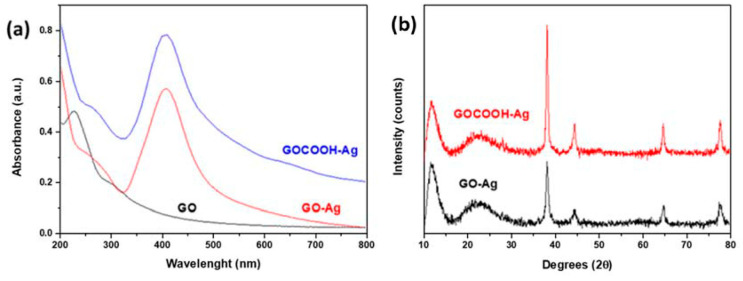
(**a**) UV–Vis absorption spectra of GO (black), GO-Ag (red) and GOCOOH-Ag (blue); (**b**) XRD spectrum for GO-Ag (black) and GOCOOH-Ag (red).

**Figure 3 nanomaterials-12-01949-f003:**
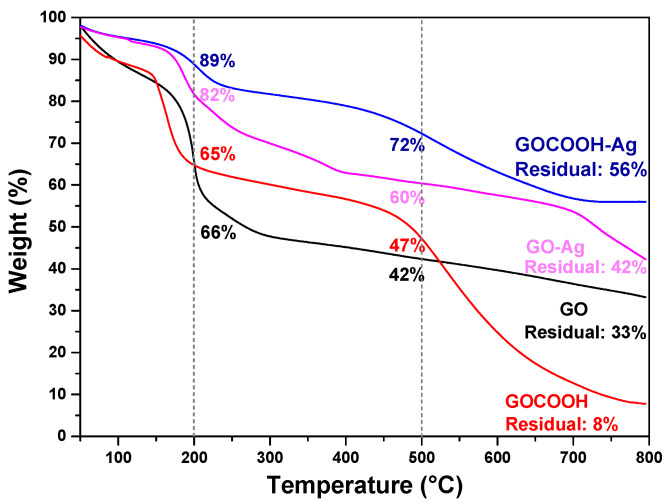
Thermogravimetric curves for GO, GOCOOH, GO-Ag, and GOCOOH-Ag.

**Figure 4 nanomaterials-12-01949-f004:**
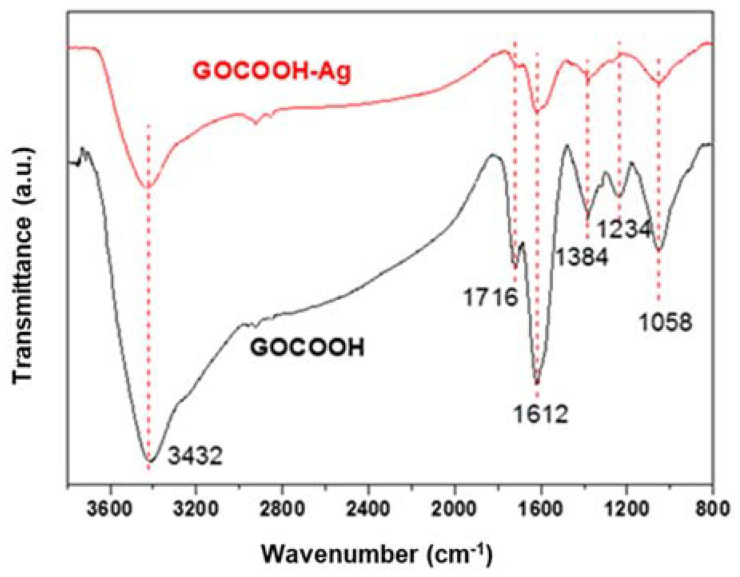
FTIR spectra for GOCOOH and GOCOOH-Ag.

**Figure 5 nanomaterials-12-01949-f005:**
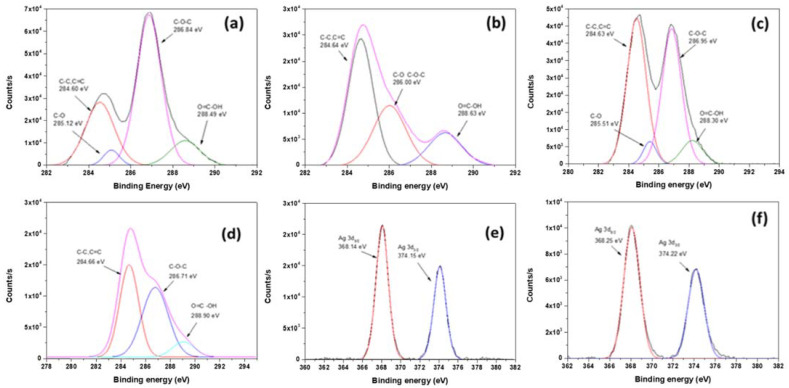
C1s XPS spectra deconvolution of (**a**) GO; (**b**) GOCOOH; (**c**) GO-Ag; (**d**) GOCOOH-Ag; and (**e**) XPS patterns of Ag3d spectra for GO-Ag and (**f**) GOCOOH-Ag.

**Figure 6 nanomaterials-12-01949-f006:**
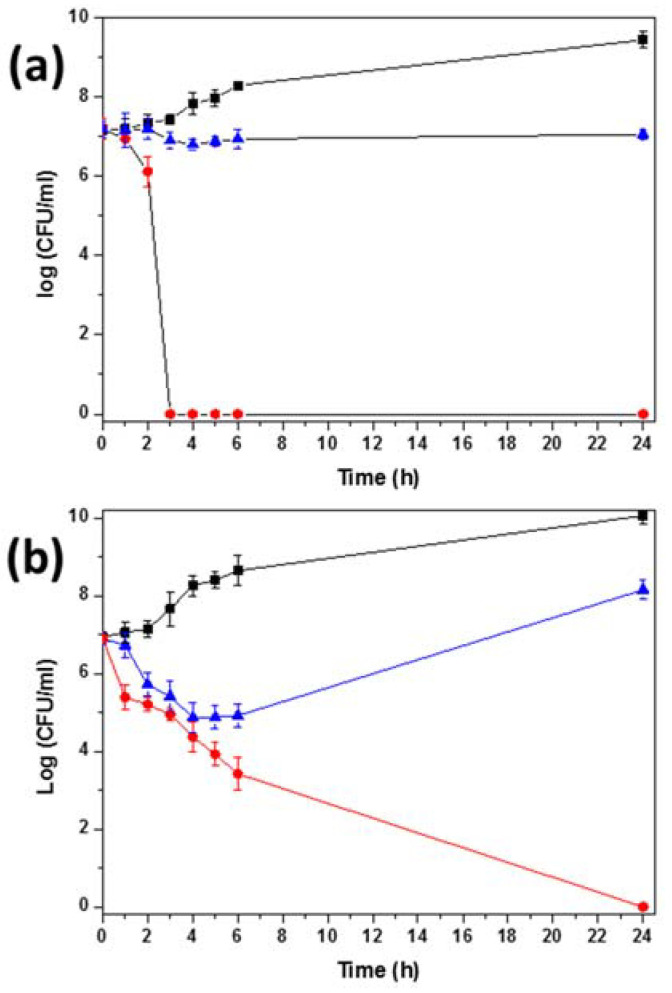
Time–kill curves of (**a**) *P. aeruginosa* (PFQ2), (**b**) *E. coli* (ATCC25922), and control (black squares) in presence of GO-Ag (blue triangles) and GOCOOH-Ag (red circles) at 12.66 µg/mL. The data were presented as means and SD of at least three independent experiments.

**Figure 7 nanomaterials-12-01949-f007:**
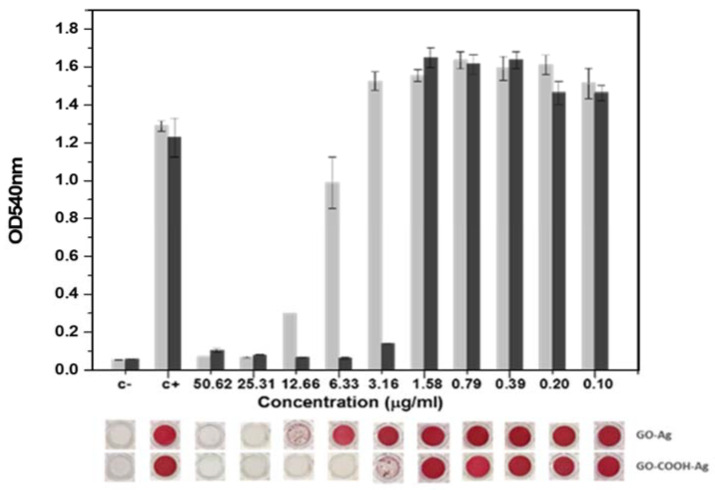
Effect of GO-Ag (light grey) and GOCOOH-Ag (dark grey) on the biofilm formation of *S. aureus* V329. Quantification (upper panel) and images (lower panel) of the biomass of a 48 h biofilm.

**Figure 8 nanomaterials-12-01949-f008:**
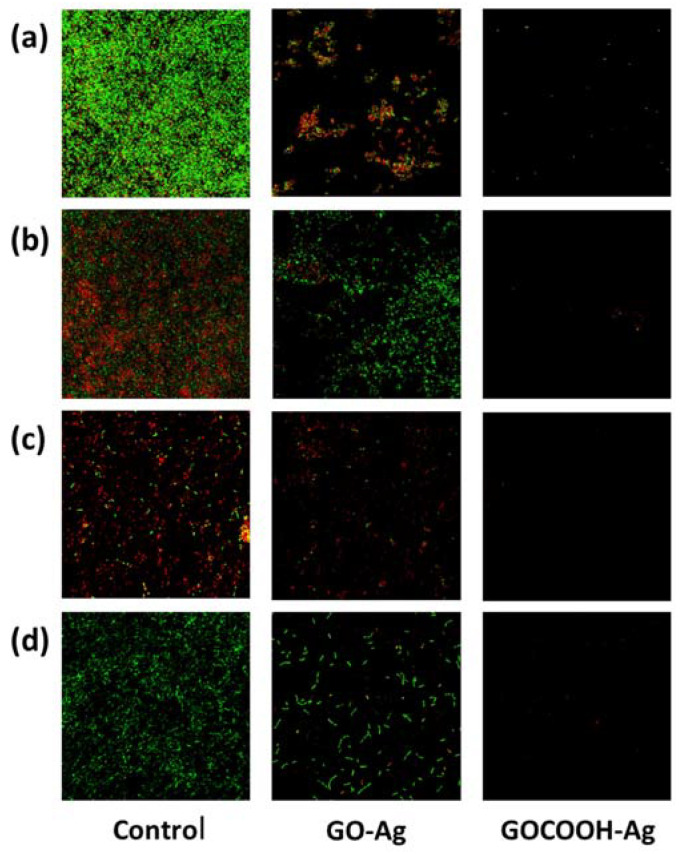
Confocal microscopy images (40x oil objective) biofilm of (**a**) *S. aureus* V329, (**b**) *S. epidermidis* RP62A, (**c**) *P. aeruginosa* PFQ2, and (**d**) *E. coli* ATCC25922. The bacteria were grown in TSBG for 48 h (control) with GO-Ag and GOCOOH-Ag at 6.33 µg/mL concentration.

**Table 1 nanomaterials-12-01949-t001:** MIC and MBC (µg/mL) of GO-Ag and GOCOOH-Ag values for all strains studied.

Species	Strains	GO-Ag			GOCOOH-Ag		
		MIC		MBC	MIC		MBC
		24 h	48 h	48 h	24 h	48 h	48 h
*S. aureus*	ATCC 25423	16.46	16.46	65.85	6.33	12.66	25.31
*S. aureus*	V329	16.46	16.46	65.85	3.16	12.66	25.31
*S. epidermidis*	ATCC32984	16.46	32.92	65.85	6.33	12.66	25.31
*S. epidermidis*	RP62A	32.92	32.92	65.85	12.66	12.66	50.62
*P. aeruginosa*	PFQ2	32.92	32.92	65.85	6.33	12.66	12.66
*E. coli*	ATCC25922	16.46	32.92	65.85	12.66	12.66	12.66

## Data Availability

Not applicable.

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
