# Peer review of "Enhanced Antibacterial Activity through Silver Nanoparticles Deposited onto Carboxylated Graphene Oxide Surface"

_nanomaterials, 2022, doi:10.3390/nano12121949_

Round 1
Reviewer 1 Report
The manuscript investigated Enhanced antibacterial activity through silver nanoparticles deposited onto carboxylated graphene oxide surface. These contents are interesting and fall within the scope of Nanomaterials. I recommend its acceptance for publication after some revisions.
1. Bacteria strains should be in italic.
2. The is no information about toxicity to eucaryotic cells
3. In antibacterial study and morphology (TEM) there is no information about control (should be free AgNPs). It is hard to compare the result where no info about free AgNPs is presented.
4. TEM experiment: mayb eadditional size distribution diagram is better to show?
5. In title one can find information about enhanced antibacterial activity of tested compounds but in article we do not know information compare to which compounds (free AgNPs ?) . Therefore, In conclusion ..Furthermore, the efficiency of AgNPs greatly improved when they were deposited onto the GOCOOH surface..... is not completly true.
Reviewer 2 Report
In the present study, the authors have focused on the synthesis and characterization of both graphene oxide and carboxylated graphene oxide-silver nanoparticles (GO-Ag and GOCOOH-Ag). The antibacterial activity and antibiofilm-forming action were tested against Gram-positive (Staphylococcus aureus and Staphylococcus epidermidis) and Gram-negative bacteria (Pseudomonas aeruginosa and Escherichia coli). Enhanced antibacterial activity for silver nanoparticles deposited onto carboxylated graphene oxide surface been observed. I recommend the manuscript to be published after major revisions.
I have some suggestion/comments/question that may contribute to the work:
- L91-95: Thermogravimetric analysis is not mentioned.
- L245-246 (Figure 1): The authors should insert the Histograms for particle size distribution and Gaussian fitting of GO-Ag and GOCOOH--Ag NPs.
- L300 (Figure 3) Mass change values for GO, GOCOOH, GO-Ag and GOCOOH-Ag under a specific temperature (for example 200, 500 C) should be inserted in Figure, as well the residual mass for each condition.
- Line 396 - (Figure 6): Time-killing curves of (a) P. aeruginosa (PFQ2) and (b) E. coli (ATCC25922) in presence of Go and GO-COOH is misisng.
- L409 - The SEM images of biofilm before and after treatments enable a more detailed analysis of the NPs effects on biofilm cells.
- L437 - According to The Live/Dead BacLight Bacterial Viability Kit, cells with a compromised membrane will stain red, whereas cells with an intact membrane strain green. As can be seen from Figure 8, the control images for (b) S. epidermidis RP62A and (c) P. aeruginosa PFQ2 are not properly, look like are not composed of viable cells. In addition, a structure specific for biofilm matrix can not be observed from confocal microscopy images for (b) S. epidermidis RP62A; (c) P. aeruginosa PFQ2 and (d) E. coli ATCC25922. The conclusion "a reduction in biofilm mass and cellular viability with respect to control was achieved" is not supported by the results in my opinion.
Reviewer 3 Report
see attached report.
Author Response
The comments of the reviewer could not be consulted since there is no attached document
Round 2
Reviewer 1 Report
I recommend this article for publishing. All comments were taken account.
Reviewer 2 Report
I recommend this article for publishing. All comments were taken account.